# OpenReview forum: "VMDT: Decoding the Trustworthiness of Video Foundation Models"
_NeurIPS.cc/2025/Datasets_and_Benchmarks_Track — NeurIPS 2025 Datasets and Benchmarks Track poster_

### Official Review · Reviewer_LiP6 · 2025-06-16

**Rating:** 5
**Confidence:** 3

**Summary:**

This paper introduces VMDT as a unified platform for evaluating T2V and V2T models across five trustworthiness dimensions: safety,
hallucination, fairness, privacy, and adversarial robustness. Through extensive evaluation of 7 T2V models and 19 V2T models using VMDT, the paper uncovers significant insights.

**Dataset Code Accessibility:**

Yes

**Dataset Code Comments:**

Repo is provided with instructions. I did not run it.

**Ethical Considerations:**

No, there are no or only very minor ethics concerns

**Final Justification:**

My concerns are addressed. I recommend *accept*.

**Limitations Weaknesses:**

I read both the main paper and briefly the supplementary. I don't see obvious limitations or weaknesses in the presented benchmark. Some questions regarding the trustworthiness of trustworthiness benchmark: (1) If a model provider does not fully agree on the proposed benchmark settings (due to e.g., political, cultural, or religious reasons) and decided to modify some of the definitions before providing benchmark results and diff, should the community (not well defined here) accept such results? (2) If a closed-source model uses a "benchmark mode" to pass the test, would there be another venue to catch the cheat?

**Strengths Contributions:**

The paper presents a thorough benchmark for V2T and T2V trustworthiness. Some insights, e.g., the lack of correlation between model size and safety, are revealing and may contribute to further development of effectively trustworthy models and GAI in general.

---

> ### Author Rebuttal · Authors · 2025-07-31
>
> We thank **Reviewer LiP6** for the insightful feedback.
>
> > **On whether the community should accept results from modified benchmarks**
>
> This is an important question. Our position is that results based on modified versions of the benchmark should not be accepted as official comparisons. Allowing individual model providers to adjust definitions risks undermining the integrity and fairness of the evaluation. While model developers are welcome to report results under modified conditions with appropriate disclaimers, such results should remain clearly distinct from the official benchmark leaderboard to preserve comparability.
>
>  > **On detecting and discouraging "benchmark mode" or cheating**
>
> We share the concern about models potentially using special "benchmark modes" to perform well only under evaluation settings. This is a legitimate challenge. One way to mitigate this is through independent third-party evaluations, which are becoming increasingly common. Many leading organizations, including frontier AI labs and non-profit auditors, are investing in this area to ensure transparency and accountability.

---

> > ### Comment · Reviewer_LiP6 · 2025-08-05
> >
> > Thank you for your response. My concerns are addressed.

---

> > > ### Author Response · Authors · 2025-08-05
> > >
> > > We sincerely appreciate your time and constructive feedback!

---

### Official Review · Reviewer_3Gb4 · 2025-06-29

**Rating:** 4
**Confidence:** 4

**Summary:**

This paper presents VMDT, the first unified benchmark for assessing the trustworthiness of VFMs, covering both text-to-video and video-to-text models. It systematically evaluates 7 T2V and 19 V2T models across five critical dimensions: safety, hallucination, fairness, privacy, and adversarial robustness, uncovering key trends and vulnerabilities.

**Dataset Code Accessibility:**

Yes

**Ethical Considerations:**

No, there are no or only very minor ethics concerns

**Final Justification:**

I sincerely appreciate the authors’ rebuttal, which helped resolve some of my confusions. I also admire the considerable effort and resources the authors have devoted to compiling such an extensive report spanning hundreds of pages. However, from the perspective of publication, I remain very hesitant about whether this paper, or perhaps more appropriately, this white paper, fits within the scope of the NeurIPS D&B track.

To be frank, the main body of the paper has little to offer; the authors have hastily compressed a large amount of content into a very limited space. After reading it, I merely found further confirmation of some of my prior understandings of the field.

Overall, I believe this is valuable work, but there is clearly a misalignment between the main paper and the supplementary material in terms of emphasis. I will therefore maintain my score.

**Limitations Weaknesses:**

1. Several evaluations, particularly in safety and hallucination, depend heavily on judgments from models like GPT-4o or Qwen2.5-VL-72B as surrogates for human evaluation. This raises concerns about potential bias or misalignment in the assessment itself, and the authors do not sufficiently analyze or justify this reliance. As for as I know, even the most powerful closed models like GPT-4o performs very average on tasks as hallucination detection.

2. The benchmark design focuses on measuring weaknesses across five dimensions but offers no clear framing for how these evaluations can translate into actionable guidance for model improvement. In particular, there is little consideration of how model designers could prioritize or balance competing dimensions (e.g., safety vs. fairness) based on these scores.

3. Given that video foundation models are rapidly evolving, the benchmark would benefit from framing its evaluations as part of an ongoing, iterative process with clear recommendations for longitudinal tracking of trustworthiness improvements. In other word, the author claimed this is a "platform", then what is the plan for future evolution given the fact the current video foundation models are still in their early stage in my view.

**Strengths Contributions:**

1. The paper makes a substantial contribution by introducing VMDT, the first unified framework that systematically evaluates both T2V and V2T models across multiple trustworthiness dimensions.

2. The authors evaluate a large and diverse set of models across well-defined tasks and scenarios.

3. I like the insights offered by this paper, especially, the analysis of how trustworthiness dimensions vary with model size and source (open vs. closed). It provides actionable insights for future model design and alignment strategies.

---

> ### Author Rebuttal · Authors · 2025-07-31
>
> We thank **Reviewer 3Gb4** for the insightful feedback.
>
> > **Reliance on LLM for evaluation**
>
> Even though human judges are the gold standard, it is prohibitively costly and limits the breadth and frequency of evaluation. For those perspectives where scalability is essential, we therefore employ LLM‑based judges. However, as the reviewer pointed out, it introduces limitations, so we took several steps to mitigate them. **We conducted careful manual verification and reported human-agreement scores for all perspectives that use LLM judges** (in the Appendix).
>
> We followed a careful process to design the LLM judges. To ensure high‑quality assessments, we experimented with several models, including GPT-4o, GPT-4o-mini, Qwen-2.5-72B, with various prompts and manually inspected their annotations based on a small set of samples. With the manual verification, we selected the best‑performing judge. As a result, agreement scores between human annotators and the chosen LLM show that our LLM judge performs comparably to human judges on our tasks (e.g., > 86 % agreement for safety). We also provide more details of our human-LLM judge agreement analyses below. We will add these to our revision.
>
> First, two human annotators independently verified the safety-LLM judges’ responses for the T2V models. Their individual agreement scores were 0.834 (Annotator 1) and 0.876 (Annotator 2), giving a mean agreement of 0.855. The overall inter-rater agreement was 0.830. The following table represents T2V safety human-LLM agreement scores by category:
>
> | Category                | Agreement |
> |-------------------------|------|
> | CSAM                    | 0.871 |
> | Intellectual Property   | 0.860 |
> | Media Types             | 0.833 |
> | NSFW                    | 0.943 |
> | Nudity & Attire         | 0.958 |
> | Offensive Content       | 0.817 |
> | Political Content       | 0.838 |
> | Sensitive Themes        | 0.882 |
> | Substance Use           | 0.726 |
> | Violence & Weapons      | 0.724 |
> | Harmful Beliefs         | 0.914 |
> | Physical harm           | 0.771 |
> | Temporal risk           | 0.929 |
>
> When we calculate a correlation between the overall safety T2V model scores annotated by LLMs and human judges, the correlation yielded 0.765 (P-value = 0.0451).
>
> Similarly, two human annotators independently verified the safety-LLM judges’ responses for the V2T models. Their individual agreement scores were 0.889 (Annotator 1) and 0.864 (Annotator 2), giving a mean agreement of 0.877. The overall inter-rater agreement was 0.846. The following table represents V2T safety human-LLM agreement scores by category:
>
> | Category                    | Agreement |
> |-----------------------------|------|
> | Fraud & Deception           | 0.865 |
> | Harassment & Bullying       | 0.917 |
> | Hateful Content             | 0.764 |
> | Physical Harm               | 0.910 |
> | Illegal & Regulated         | 0.858 |
> | Adult Content               | 0.896 |
>
> When we calculate a correlation between the overall safety V2T model scores annotated by LLMs and human judges, the correlation yielded 0.969 (P-value = 9.219e-12).
>
> For hallucination T2V, we additionally added a second annotator for robustness. When we calculated a correlation between scores from human annotators and scores from the LLM judge, the correlation values yielded as follows: The average human scores vs. the LLM judge is a correlation of 0.765 (P-value=6.208e-219); for annotator 1 vs the LLM judge is 0.741 (P-value=1.883e-198), and annotator 2 vs the LLM judge is 0.715 (P-value=1.893e-178). For comparison, the correlation between two human annotators themselves is 0.812 (P-value=4.673e-267). Because the human-LLM correlations are close to the inter-annotator correlation, the LLM judge shows agreement comparable to a human judge.
>
> We also report a correlation of the average model scores obtained by human judges and our LLM judge: 0.9705 (P-value=0.0003). The correlation of the average model scores for each scenario is below:
> | Scenario              | Correlation | P-value  |
> |--------------------|-----------------|----------|
> | Misleading         | 0.982          | 0.0001   |
> | NaturalSelection   | 0.892          | 0.0070   |
> | CoOccurrence       | 0.964          | 0.0005   |
> | OCR                | 0.986          | 0.0000   |
> | Counterfactual     | 0.943          | 0.0015   |
> | Distraction     | 0.714          | 0.0714   |
> | Temporal           | 0.758          | 0.0806   |
>
> As a result, our human-LLM judge agreement analysis indicates that our LLM judges are comparable to human judges. Moreover, our automated evaluation scripts make it easy to swap in stronger judge models in the future as they become available.
>
> > **A lack of discussion on strategies or guidance for developers**
>
> We agree that converting evaluation scores into actionable guidance is crucial. In the revision, we therefore (i) add a dedicated subsection (“How to enhance trustworthiness”) that links each trustworthiness dimension to actionable guidance. In the subsection, we will discuss the following:
> - Safety: Strengthening safety in VFMs requires a multi‑layered strategy that combines rigorous input‑ and output‑filtering pipelines, alignment processes such as RLHF or constitutional RL calibrated to a domain‑specific risk taxonomy, and continuous red‑team stress‑testing to surface emerging failure modes. The pronounced gap we observe between open‑source and closed‑source models on VMDT’s safety axis indicates that many open models significantly lack these defense layers; our findings therefore underscore the urgency for the open‑source community to embrace robust safety‑engineering practices.
> - Hallucination: Mitigating hallucinations has long been challenging. In particular, we find that VFMs are especially susceptible to video‑specific failure modes, such as temporal inconsistencies. However, they also retain many of the same problems as image-focused models, having difficulties with complex tasks like spatial understanding and counting. To address hallucination, better data needs to be collected targeting a more diverse set of scenarios. While open and closed source models are similar in the V2T setting, the difference in T2V performance suggests large variations in data quantity and quality that need to be addressed. Additionally, techniques enabling better consistency between frames should be explored.
> - Fairness: Effective bias mitigation can start with curating training data that deliberately balance sensitive attributes (gender, ethnicity, and age), while supplementing gaps through synthetic resampling and counterfactual editing. Complementary, lightweight activation‑level interventions have recently proved useful for mitigating residual demographic biases. Our evaluation, however, also reveals a counter‑risk of “overfairness.” Mitigations should therefore be conducted incrementally and tracked with slice‑wise metrics that surface both residual bias and over‑correction, enabling practitioners to reach a calibrated trade‑off that matches the ethical and functional requirements of their deployment context.
> - Privacy: Protecting user or proprietary content in VFMs involves a spectrum of privacy‑preserving learning paradigms such as differential privacy and federated or split‑learning pipelines. Complementary data‑centric tactics (e.g., blurring faces or license numbers) can lower the risk of identity leakage.
> - Adversarial robustness: Ensuring robustness against adversarial attacks requires a multi-faceted approach. First, lightweight input filtering can catch and denoise adversarially perturbed inputs. Second, incorporating adversarial data during training can improve robustness. Finally, as observed in our evaluation, scaling can be an effective strategy; larger models tend to be more robust than smaller models.
>
> > **Recommendations and guidance for longitudinal tracking considering the early stage of video foundation models**
>
> Thank you for highlighting this need. We plan to launch a community‑driven, continuously updated public leaderboard that accepts VMDT evaluation results from developers and researchers. This will let us track progress over time, hold the field collectively accountable, and guide steady improvements in safety, hallucination, fairness, privacy, and robustness.

---

> > ### Author Response · Authors · 2025-08-04
> >
> > Dear Reviewer 3Gb4,
> >
> > Thank you again for your valuable feedback! In our rebuttal, we have carefully addressed all the points you raised.
> > Given that the discussion phase deadline is approaching, we would be grateful if you could take a moment to review our response.
> > We welcome any additional feedback you might have.
> >
> > Best,
> > Authors

---

### Official Review · Reviewer_yBbQ · 2025-07-02

**Rating:** 6
**Confidence:** 4

**Summary:**

This paper introduces the first unified platform for evaluating the trustworthiness of T2V and V2T models, named VMDT. The platform assesses models across five key dimensions: safety, hallucination, fairness, privacy, and adversarial robustness. Through a comprehensive evaluation of 7 T2V models and 19 V2T models, the study finds that all open-source T2V models fail to identify harmful prompts and often generate harmful content. These models also exhibit higher unfairness compared to image-based models. For V2T models, unfairness and privacy risks increase with model scale, while hallucination and adversarial robustness improve slightly but remain low overall. Furthermore, no clear correlation is found between model size and safety. This study highlights the urgent need to develop more trustworthy video foundation models, and VMDT offers a systematic framework for measuring and tracking progress in this area.

**Dataset Code Accessibility:**

Yes

**Dataset Code Comments:**

The author provides access for obtaining the dataset, and the dataset has a good structure.

**Ethical Comments:**

I don't think this paper involves ethical issues.

**Ethical Considerations:**

No, there are no or only very minor ethics concerns

**Final Justification:**

The author's reply has solved my all concerns.

**Limitations Weaknesses:**

1. Regarding insights into improving video foundation models, although the paper evaluates existing open-source and closed-source models on trustworthiness, it lacks a discussion on strategies to enhance trustworthiness. Including some potential approaches would help advance the field.

2. The paper barely discusses the performance of the involved video foundation models on text-to-video and video-to-text tasks. Ignoring performance entirely when evaluating trustworthiness may introduce bias. It is recommended to include a discussion on model performance in appropriate sections.

3. In Section 2, when calculating the Bypass Rate, it is unclear whether the evaluated video foundation models incorporate any defense strategies such as prompt filters or safety checkers, or if the evaluation relies solely on the raw models. In real-world commercial applications, service providers usually employ safety mechanisms to prevent the generation of NSFW content.

4. Format issue: The Datasets and Benchmarks Track uses single-blind review, and the main text of the paper should include author information.

**Strengths Contributions:**

1. The authors focus on the trustworthiness of video foundation models, which is beneficial for advancing the community's understanding of this important issue.

2. The authors propose VMDT, the first unified platform for evaluating the trustworthiness of both T2V and V2T models, filling a significant gap.

3. The VMDT platform evaluates five key dimensions—safety, hallucination, fairness, privacy, and adversarial robustness—covering the main aspects of trustworthiness.

4. For each dimension, the authors construct datasets and conduct evaluations and analyses, showing substantial effort in implementation.

---

> ### Author Rebuttal · Authors · 2025-07-31
>
> We thank **Reviewer yBbQ** for the insightful feedback.
>
> > **A lack of discussion on strategies to enhance trustworthiness**
>
> We agree that converting evaluation scores into actionable guidance is crucial. In the revision, we therefore (i) add a dedicated subsection (“How to enhance trustworthiness”) that links each trustworthiness dimension to actionable guidance. In the subsection, we will discuss the following:
> - Safety: Strengthening safety in VFMs requires a multi‑layered strategy that combines rigorous input‑ and output‑filtering pipelines, alignment processes such as RLHF or constitutional RL calibrated to a domain‑specific risk taxonomy, and continuous red‑team stress‑testing to surface emerging failure modes. The pronounced gap we observe between open‑source and closed‑source models on VMDT’s safety axis indicates that many open models significantly lack these defense layers; our findings therefore underscore the urgency for the open‑source community to embrace robust safety‑engineering practices.
> - Hallucination: Mitigating hallucinations has long been challenging. In particular, we find that VFMs are especially susceptible to video‑specific failure modes, such as temporal inconsistencies. However, they also retain many of the same problems as image-focused models, having difficulties with complex tasks like spatial understanding and counting. To address hallucination, better data needs to be collected targeting a more diverse set of scenarios. While open and closed source models are similar in the V2T setting, the difference in T2V performance suggests large variations in data quantity and quality that need to be addressed. Additionally, techniques enabling better consistency between frames should be explored.
> - Fairness: Effective bias mitigation can start with curating training data that deliberately balance sensitive attributes (gender, ethnicity, and age), while supplementing gaps through synthetic resampling and counterfactual editing. Complementary, lightweight activation‑level interventions have recently proved useful for mitigating residual demographic biases. Our evaluation, however, also reveals a counter‑risk of “overfairness.” Mitigations should therefore be conducted incrementally and tracked with slice‑wise metrics that surface both residual bias and over‑correction, enabling practitioners to reach a calibrated trade‑off that matches the ethical and functional requirements of their deployment context.
> - Privacy: Protecting user or proprietary content in VFMs involves a spectrum of privacy‑preserving learning paradigms such as differential privacy and federated or split‑learning pipelines. Complementary data‑centric tactics (e.g., blurring faces or license numbers) can lower the risk of identity leakage.
> - Adversarial robustness: Ensuring robustness against adversarial attacks requires a multi-faceted approach. First, lightweight input filtering can catch and denoise adversarially perturbed inputs. Second, incorporating adversarial data during training can improve robustness. Finally, as observed in our evaluation, scaling can be an effective strategy; larger models tend to be more robust than smaller models.
>
> > **A lack of discussion on their performance**
>
> We fully agree that ignoring model capability can bias model assessments. In the revision, we will add a discussion to Section 7 that links our size‑dependent findings to capability. Our results indicate that stronger models exhibit fewer hallucinations and greater adversarial robustness, yet they also amplify demographic stereotypes and preferences and increase privacy leakage—likely because enhanced capacity extracts both helpful and sensitive patterns from the data. This tension shows that no single model configuration currently optimizes every trustworthiness dimension, underscoring the continued need for targeted mitigation efforts in VFM trustworthiness.
>
> > **Clarification for Bypass Rate**
>
> We evaluated each model exactly as an end user would: closed‑source systems were accessed through their public APIs, while open‑source systems were run from the official Hugging Face checkpoints. For proprietary models, we cannot inspect or control any internal safety layers (e.g., prompt filters or policy checkers); our results therefore reflect whatever defences are active behind the public interface. In practice, OpenAI’s models returned refusal messages rather than API errors. Our Bypass Rate counts both API errors and explicit refusals, and considers a request successfully bypassed only when the model produces a substantive non‑refusal, non‑error output.

---

> > ### Comment · Reviewer_yBbQ · 2025-08-06
> >
> > The author's reply has solved my problem. I will increase my score.

---

> > > ### Author Response · Authors · 2025-08-06
> > >
> > > We sincerely appreciate your time and constructive feedback again, and we will ensure that our revision addresses the points discussed in the rebuttal.

---

### Official Review · Reviewer_P3a5 · 2025-07-02

**Rating:** 4
**Confidence:** 4

**Summary:**

This paper introduces VMDT (Video Modal DecodingTrust), the first comprehensive benchmark platform for evaluating the trustworthiness of video foundation models (VFMs). The benchmark assesses both text-to-video (T2V) and video-to-text (V2T) models across five critical dimensions: safety, hallucination, fairness, privacy, and adversarial robustness. Through extensive evaluation of 7 T2V models and 19 V2T models, the authors uncover important insights including size-dependent trends, safety gaps between open and closed-source models, and various model vulnerabilities.

**Dataset Code Accessibility:**

Yes

**Ethical Considerations:**

No, there are no or only very minor ethics concerns

**Final Justification:**

I appreciate the authors' thorough rebuttal, which has effectively addressed the majority of my concerns. I have raised my score accordingly.

**Limitations Weaknesses:**

1.	The paper lacks innovation in developing video-specific evaluation metrics that could better capture temporal dynamics, motion coherence, or video-unique safety risks.
2.	The paper relies heavily on GPT-4o for evaluation across multiple dimensions without thoroughly investigating the potential biases. For example, GPT-4o might have systematic biases in judging certain types of content or demographics that could skew the trustworthiness scores.
3.	The paper presents extensive empirical results but does not explain why certain patterns emerge (e.g., why safety shows no correlation with model size while other dimensions do). Beyond identifying problems, the paper offers minimal guidance on how to improve model trustworthiness.
4.	The paper doesn't adequately address potential data contamination issues, particularly for closed-source models that might have been trained on similar safety benchmarks or evaluation datasets.

**Strengths Contributions:**

The benchmark fully covers five essential trustworthiness aspects with carefully curated datasets, providing both breadth and depth in evaluation scenarios that capture video-specific challenges like temporal risks. boosting practical guidance for model development.

---

> ### Author Rebuttal · Authors · 2025-07-31
>
> We thank **Reviewer P3a5** for the insightful feedback.
>
> > **A lack of innovation for video‑specific characteristics and video‑unique safety risks**
>
> We respectfully disagree that the benchmark is “non‑innovative” for video‑specific evaluation. Our benchmark design was driven by characteristics that do not appear in image or text settings, and we introduced new tasks, metrics, and datasets accordingly:
>
> - For safety, we moved beyond static-frame checks by injecting temporal and physical‑hazard scenarios (e.g., harmful content emerging only when viewed in sequence, content that may trigger photosensitive epilepsy through rapid flashing or strobe effects). Our evaluation reveals that current VFMs struggle disproportionately with such hazards.
> - For hallucination, we curated and generated clips that stress temporal consistency, motion coherence, and multi‑scene reasoning. Moreover, we developed a metric (Section D.1 in Appendix) that can capture hallucination levels in a more fine-grained way (e.g., how many frames generated by T2V models have hallucinations).
> - Our fairness evaluation highlights modality‑specific bias: T2V systems exhibit stronger male/White preference than T2I, and our “overkill fairness’’ videos for V2T show that a model (e.g., GPT‑4o) becomes markedly more incorrect when demographic cues evolve across frames (Appendix E.2.3).
> - Our privacy evaluation also identifies the disparity between video modality and image modality models. Even though T2V shows significantly less data memorization risk than T2I, V2T exhibits increasing privacy leakage as model capacity grows, likely because stronger cross‑modal mappings recover more fine‑grained identifying details from the training corpus.
> - Finally, the adversarial robustness set employs greedy, genetic, gradient‑based, and FMM‑Attack perturbations tailored to temporal data. We filtered adversarial samples with surrogate models to ensure high attack success and semantic plausibility, a pipeline that, to our knowledge, has not been applied to large‑scale VFM testing before.
>
> > **Reliance on GPT-4o for evaluation**
>
> Even though human judges are the gold standard, it is prohibitively costly and limits the breadth and frequency of evaluation. For those perspectives where scalability is essential, we therefore employ LLM‑based judges. However, as the reviewer pointed out, it introduces limitations, so we took several steps to mitigate them. **We conducted careful manual verification and reported human-agreement scores for all perspectives that use LLM judges** (in the Appendix).
>
> We followed a careful process to design the LLM judges. To ensure high‑quality assessments, we experimented with several models, including GPT-4o, GPT-4o-mini, Qwen-2.5-72B, with various prompts and manually inspected their annotations based on a small set of samples. With the manual verification, we selected the best‑performing judge. As a result, agreement scores between human annotators and the chosen LLM show that our LLM judge performs comparably to human judges on our tasks (e.g., > 86 % agreement for safety). We also provide more details of our human-LLM judge agreement analyses below. We will add these to our revision.
>
> First, two human annotators independently verified the safety-LLM judges’ responses for the T2V models. Their individual agreement scores were 0.834 (Annotator 1) and 0.876 (Annotator 2), giving a mean agreement of 0.855. The overall inter-rater agreement was 0.830. The following table represents T2V safety human-LLM agreement scores by category:
>
> | Category                | Agreement |
> |-------------------------|------|
> | CSAM                    | 0.871 |
> | Intellectual Property   | 0.860 |
> | Media Types             | 0.833 |
> | NSFW                    | 0.943 |
> | Nudity & Attire         | 0.958 |
> | Offensive Content       | 0.817 |
> | Political Content       | 0.838 |
> | Sensitive Themes        | 0.882 |
> | Substance Use           | 0.726 |
> | Violence & Weapons      | 0.724 |
> | Harmful Beliefs         | 0.914 |
> | Physical harm           | 0.771 |
> | Temporal risk           | 0.929 |
>
>
> When we calculate a correlation between the overall safety T2V model scores annotated by LLMs and human judges, the correlation yielded 0.765 (P-value = 0.0451).
>
> Similarly, two human annotators independently verified the safety-LLM judges’ responses for the V2T models. Their individual agreement scores were 0.889 (Annotator 1) and 0.864 (Annotator 2), giving a mean agreement of 0.877. The overall inter-rater agreement was 0.846. The following table represents V2T safety human-LLM agreement scores by category:
>
> | Category                    | Agreement |
> |-----------------------------|------|
> | Fraud & Deception           | 0.865 |
> | Harassment & Bullying       | 0.917 |
> | Hateful Content             | 0.764 |
> | Physical Harm               | 0.910 |
> | Illegal & Regulated         | 0.858 |
> | Adult Content               | 0.896 |
>
> When we calculate a correlation between the overall safety V2T model scores annotated by LLMs and human judges, the correlation yielded 0.969 (P-value = 9.219e-12).
>
> For hallucination T2V, we additionally added a second annotator for robustness. When we calculated a correlation between scores from human annotators and scores from the LLM judge, the correlation values yielded as follows: The average human scores vs. the LLM judge is a correlation of 0.765 (P-value=6.208e-219); for annotator 1 vs the LLM judge is 0.741 (P-value=1.883e-198), and annotator 2 vs the LLM judge is 0.715 (P-value=1.893e-178). For comparison, the correlation between two human annotators themselves is 0.812 (P-value=4.673e-267). Because the human-LLM correlations are close to the inter-annotator correlation, the LLM judge shows agreement comparable to a human judge.
>
> We also report a correlation of the average model scores obtained by human judges and our LLM judge: 0.9705 (P-value=0.0003). The correlation of the average model scores for each scenario is below:
> | Scenario              | Correlation | P-value  |
> |--------------------|-----------------|----------|
> | Misleading         | 0.982          | 0.0001   |
> | NaturalSelection   | 0.892          | 0.0070   |
> | CoOccurrence       | 0.964          | 0.0005   |
> | OCR                | 0.986          | 0.0000   |
> | Counterfactual     | 0.943          | 0.0015   |
> | Distraction     | 0.714          | 0.0714   |
> | Temporal           | 0.758          | 0.0806   |
>
> As a result, our human-LLM judge agreement analysis indicates that our LLM judges are comparable to human judges. Moreover, our automated evaluation scripts make it easy to swap in stronger judge models in the future as they become available.
>
> > **Limited explanation of observed patterns and minimal guidance on improvement**
>
> We appreciate the reviewer’s request for deeper causal insight behind the empirical trends. Our contribution lies in surfacing systematic patterns; a definitive causal analysis would require extensive interpretability studies and therefore remains as future work. Nonetheless, we will add a discussion on some hypotheses in §7. First, we suspect that larger‑capacity models show fewer hallucinations and stronger adversarial resilience because broader temporal context and richer latent representations enable greater consistency. The same capacity, however, can amplify demographic biases and privacy leakage, likely because the models more accurately recover subtle demographic cues, better reflect training data, and memorize personal details embedded in their training data. Safety’s near‑zero correlation with scale suggests that current safety performance is governed mainly by post‑training alignment stages (e.g., RLHF, policy fine‑tuning) rather than raw parameter count.
>
> We will also add concrete mitigation avenues to help practitioners translate VMDT findings into actionable improvements (please see the response to Reviewer yBbQ's first question).
>
> > **Potential data contamination issues**
>
> With the exception of the hallucination-V2T suite only for the natural selection scenario and the privacy‑T2V suite, **every prompt in VMDT was newly written for this work**. For several tasks, we reused publicly available video clips from earlier datasets, but we paired each clip with our own, previously unseen textual queries or descriptions, so the combined video‑prompt pairs could not have appeared in any pre‑training set. The hallucination V2T natural‑selection scenario aims to measure how much hallucination occurs on naturally difficult prompts without any modifications. To do this, we use existing prompts drawn from a variety of datasets; in other scenarios, however, the prompts are transformed or newly created (e.g., for OCR). The privacy‑T2V benchmark is also one intentional overlap: we recycled prompts from WebVid‑10M, which is a popular training data, to test memorization behaviour in models. Moreover, there are still very few publicly released trustworthiness benchmarks for video, and therefore accidental overlap elsewhere is improbable. Finally, if contamination were the main driver of our results, we would expect conspicuous outliers, yet the scores across both closed‑ and open‑source models remain similar (see Section 7 and Table 54). Even though closed-source models scored highly compared to open-source V2T models for the safety part, the results align with many prior evaluation results for LLM and I2T models.

---

> > ### Author Response · Authors · 2025-08-04
> >
> > Dear Reviewer P3a5,
> >
> > Thank you again for your valuable feedback! In our rebuttal, we have carefully addressed all the points you raised.
> > Given that the discussion phase deadline is approaching, we would be grateful if you could take a moment to review our response.
> > We welcome any additional feedback you might have.
> >
> > Best,
> > Authors

---

> > > ### Comment · Reviewer_P3a5 · 2025-08-05
> > >
> > > I appreciate the authors' thorough rebuttal, which has effectively addressed the majority of my concerns. I have raised my score accordingly.

---

> > > > ### Author Response · Authors · 2025-08-05
> > > >
> > > > We sincerely appreciate your time and constructive feedback, and we will ensure that our revision addresses the points discussed in the rebuttal.

---

### Decision · Program_Chairs · 2025-09-18

**Decision:**

Accept (poster)

**Comment:**

This paper introduces a benchmark for assessing the trustworthiness of video foundation models in aspects like safety, hallucination, fairness, privacy, and adversarial robustness. Evaluation is comprehensive with 20+ models. Reviewers generally think the benchmark is timely, systematic, and impactful, with broad coverage and good insights. For example, it is found that the T2V models would generate harmful videos and fail to filter harmful user queries. It is also observed that these T2V models exhibit higher levels of unfairness compared to image modality models. While concerns were raised about reliance on LLM judges and scope, authors' rebuttal look convincing. Overall, the work is valuable in evaluating VFMs and offers a valuable resource for the community. AC recommends accept.

===== FINAL UPDATE FROM DB Track PCs ====

The final decision for this paper has been taken by the program chairs after consultation with the SACs. All Senior Area Chairs have ranked papers according to the feedback from the AC during the review process. We decided to leave the original meta-review to reflect the opinion of the AC in light of the initial discussions with reviewers and SAC.